# UAV Noise Emission—A Combined Experimental and Numerical Assessment

Kai Cussen [†], Simone Garruccio [†] and John Kennedy *[,†]

Department of Mechanical, Manufacturing and Biomedical Engineering, Trinity College Dublin, University of Dublin, D02 PN40 Dublin, Ireland; kcussen@tcd.ie (K.C.); garruccs@tcd.ie (S.G.)
* Correspondence: jkenned5@tcd.ie
† These authors contributed equally to this work.

**Abstract:** Noise emission will be a significant obstacle to the widespread uptake of unmanned aerial vehicles or UAVs. The assessment and mitigation of UAV noise will require validated modelling approaches. The European Union has recently mandated an UAV sound power measurement procedure based on a procedure for measuring machinery or equipment. It is not clear if this legally mandated noise assessment will provide useful data for environmental noise modelling of UAVs. This research aimed to determine the sound power level of a UAV according to the legally mandated ISO 3744 and to investigate the suitability of commercial implementations of ISO 9613 for modelling noise emission from UAVs. A class C1 UAV was used for the investigation which also included controlled flyover tests. Several different operating conditions were measured and modelled and the results compared. The small scale UAV used had a sound power of 86.8 dB (A) and modelled flyover tests agreed with experimental values within ±2.1 decibels at distances up to 30 m and within angles of 45–90° of the receiver. The validated model was then used for a case study of UAV noise emission in an urban setting. The model demonstrated the potential for UAV noise emission to significantly exceed urban background noise levels by up to 10 dB. It was found that flight altitude relative to building height had a significant impact on the number of allowable UAV operations within WHO $L_{DEN}$ guidelines.

**Keywords:** UAV; environmental noise; sound power level





## 1. Introduction

UAV usage has expanded in recent years due to their range of applications from package delivery, agriculture, construction, and photography. Companies such as Amazon and Alphabet have begun trials using UAVs to deliver packages to their customers [1,2] and smaller businesses such as pharmacies have also trialled medication delivery using UAVs [3]. This comes with many advantages such as smaller waiting times for deliveries, and lower Carbon dioxide emissions than conventional delivery methods [4]. With an increase of UAVs in the skies, there is likely to be an increase in environmental noise, which could cause annoyance among residents living under or near regularly used flight paths [5].

The World Health Organization have noted that noise is a "growing concern". Several adverse health effects such as sleep disorders with awakenings [6], cognitive impairment [7,8], hypertension ischemic heart disease [9,10], diastolic blood pressure [11], reduction of working performance [12,13] and annoyance [14] have been linked to an over exposure [15]. A study by Airbus found that noise is one of the top concerns among the public in relation to increased UAV usage [16], and it has been suggested that in environments with less road traffic, the annoyance of UAV noise is significantly higher [17]. At time of writing, there are no universal standards dealing with this issue. To reduce the annoyance and health effects caused by UAV noise, some state bodies, such as the European Union and Australia's Department of Infrastructure have begun to implement regulations

tackling the issue [18,19] (Australia's Department of Infrastructure also recently reviewed their regulations pertaining to UAV noise [20]).

The EU have recently introduced regulations covering testing standards for different classes of UAVs, as well as setting maximum sound power levels that have been assigned to certain classes of UAV [18,21]. (These regulations were also retained in UK law [22].) Parts 13–15 of these regulations specifically addresses regulation and testing requirements for UAV noise. The method for determining the A-weighted sound power level for different classes of UAVs is prescribed and testing follows the ISO 3744:2010 standards. They require the UAV to be tested under hovering conditions at its Maximum Take Off Mass (MTOM), above one reflecting surface and sufficiently far away from any other reflecting surface. The testing is carried out in a hemispherical measurement surface and the number and position of the microphones used is as described in annex F of EN ISO 3744:2010. These regulations introduce the requirement for the guaranteed sound power level to be represented in a pictogram on the UAV shown in Figure 1. They also describe the maximum allowable sound power level of the UAV depending on its class and includes the requirements for a reduction in maximum sound power level from 2 years after the document came into force, and from 4 years after the document came into force. The legally required pictogram only requires a A-weighted total sound power level but the ISO 3744 procedure will easily enable frequency dependent data to be captured as a function of third octave band.

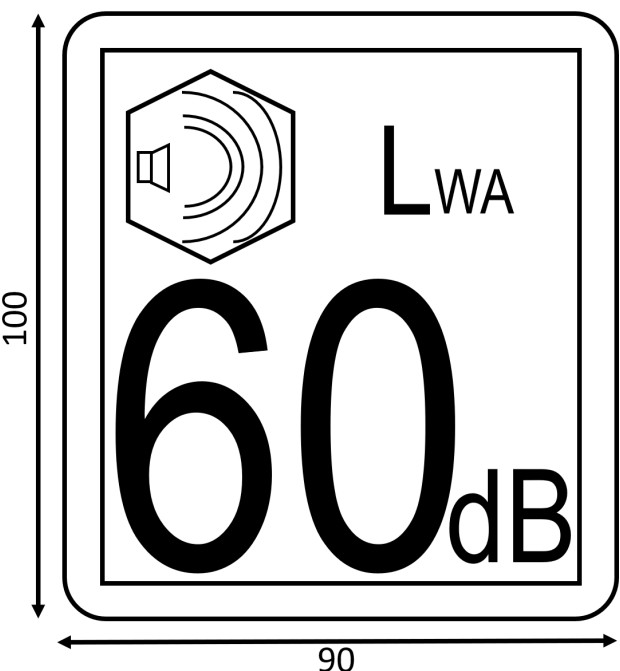

**Figure 1.** EU regulated pictogram providing indication of the guaranteed sound power level units in mm [18].

The department of infrastructure in Australia have released the 'Air Navigation (Aircraft Noise) Regulations 2018'. They mention requirements for 'Approval for other aircraft to which no standards apply'. These include commercial UAVs and involve applying to the secretary for approval to commence in air navigation. These standards are not specifically targeting UAVs and a review of the regulations was undertaken specifically to understand the need for UAV noise regulations [19]. Australian legislation aims to regulate noise from large commercial UAVs flying in urban areas, however regulations exclude personal UAVs or UAVs used by emergency services. The review recommended interim noise regulations specifically for commercial UAVs focusing on operations likely to have significant noise impacts, an interim eVTOL (electric vertical takeoff and landing) noise management framework that considers noise impacts during take-off and landing, use

of noise forecasting tools to communicate expected noise impacts to local communities and planning authorities, and introducing necessary operating limits to ensure acceptable noise levels within the community. This framework is suggested to remain in place until an enduring noise policy framework for the future can be established.

Several key findings appear in currently published works surrounding the issue of UAV noise. Torija et al. have found that public perception of UAV noise varies significantly in different soundscapes. It was found that in soundscapes with little road traffic noise, the perceived annoyance of UAV noise was 6.4 times higher than in soundscapes that are highly impacted by road noise [17]. The paper concludes by suggesting that concentrating UAV flight paths along busy roads could assist in mitigating the perceived annoyance of UAV noise. This finding was echoed by Palmer et al. [23], here a rise of 5–6 dB of noise was shown to make "sporadic complaints" become "widespread". The noise emitted from UAVs has been found to not qualitatively resemble aircraft noise and has also been reported as more annoying than noise from traditional road vehicles and aircraft of the same 'loudness' level [24].

Noise emission from UAVs is a complex issue. It is generally accepted that UAV noise is composed of two parts, tonal noise, and broadband noise. Emitted sound has been shown to be related to the number of propeller blades, and the number of revolutions per second of the blades [25]. Tonal noise has been found to stem from the noise emitted at the BPF (blade passage frequency) [26] (for contra rotating blades). This paper also notes that noise is caused by the brushless motors typically used in UAVs (at harmonics of the rotational speed) and turbulence. Broadband noise can stem from the turbulence of the incoming flow, flow separation at the aerofoil, and the turbulent boundary layer at the leading edge of the aerofoils [27]. The predominant source of noise has been seen to vary depending on the frequency being examined. Jordan et al. found that propeller noise is a major source of noise at lower frequencies, while nonrotor noise dominates at higher frequencies [28], increased pitch angle has also been linked with higher broadband and tonal noise due to increased interaction between the flow and the trailing edge and tip of the aerofoil [29].

In summary, as this is a relatively new area of interest, there are currently few countries with legislation specifically regarding noise emission from UAVs. However, as the EU and Australia have already demonstrated, it is likely that more legislation to deal with the issue will be created soon. The predominant sources of noise from UAVs are the motors, turbulence, flow separation and harmonics of the UAV's BPF. From the currently available research, it appears that the soundscape the UAV is operating in will have a significant effect on its overall perceived annoyance. Due to the new EU legislation an acoustic study of every UAV operated in the European Union is about to occur, measured using ISO 3744. This will lead to a huge number of UAV noise measurements across many countries but prior to this work it was not clear if this would provide any useful information for UAV noise mapping. Therefore this work aims to take a sample UAV through the full process from sound power measurement, through to fly over validation and then a case study relevant for environmental noise problems.

## 2. Numerical Method

The modelling was carried out on commercial software iNoise which follows the calculation methods of ISO 9613. These methods describe how to calculate sound attenuation during outdoor propagation to predict sound levels at various locations and distances from the source(s) [30,31]. Part one of ISO 9613 deals with sound attenuation due to atmospheric absorption, while part two deals with the general calculation method to predict the levels of environmental noise and equivalent continuous A-weighted sound pressure levels.

The sound emitted from the UAV is likely direction dependent. The directivity of the source can be accounted for using a directivity correction which will take into account the amount the sound deviates from an omnidirectional source. This is also described in part 2

of ISO 9613. The equivalent continuous downwind octave band sound pressure level is therefore calculated by Equation (1).

$$L_{ft}(DW) = L_W + D_C - A \tag{1}$$

where $L_W$ is the sound power level of the UAV, $D_C$ is the directivity correction and $A$ is the attenuation. ISO 9613 is the standard calculation method for calculation of outdoor propagation of sound. There are many commercially available implementations and iNoise was the chosen software for the purposes of this research. This software meets the verification problems specified within ISO 17534 to within $\pm$0.05 dB.

The software being used requires several inputs to calculate SPL values. The sound power level of the source (in this case, the UAV) is required. Third octave bands were used in this research, however the calculation method is also applicable for octave bands. Atmospheric conditions such as temperature, pressure and humidity are also required. There is also an option to include the directivity of the sound source, either 2D or 3D directivity can be used. The altitude of the sound source can be inputted, as well as ground factors for surrounding terrain.

### 3. Measurement of UAV Sound Power

In order to allow comparisons with other commercial UAVs the performance of the HS720 propeller blades were tested in an experimental rig designed to measure thrust vs. RPM. A HX711 1 kg load cell was used as the thrust measurement device. A motor stand was designed in such a way as to accommodate the load cell while connecting it to the motor & base plate. A KY-008 650 nm laser emitter/receiver pair was used to measure the RPM of the propeller. The devices were all connected using an Arduino UNO board. The results of the thrust testing of the propeller blades can be seen in Figure 2.

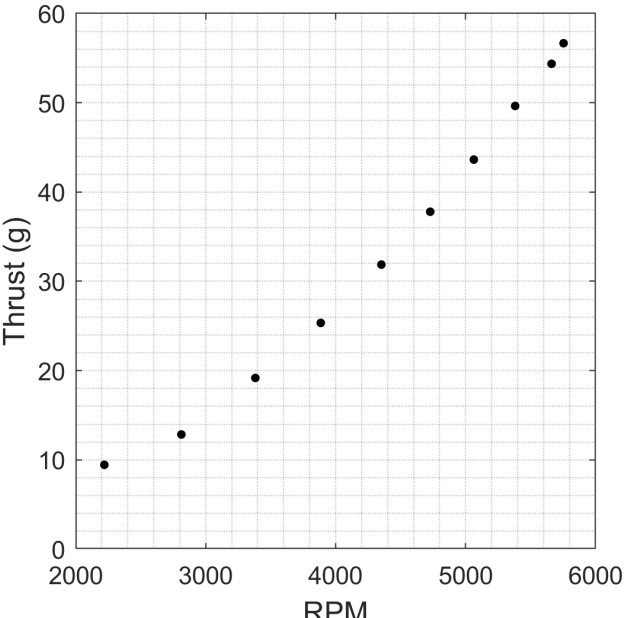

**Figure 2.** Thrust vs. RPM.

As the software and calculation standard have not yet been approved for the calculation of UAV noise, results from field tests were compared against an the software model of the test to asses whether the method is capable of accurately calculating the SPL due to a UAV source. The first stage of this process was to estimate the UAV sound power according to ISO 3744:2010, the standard specified by EU UAV legislation. This standard is now legally mandated for the measurement of UAV sound power. This may be of some concern to acousticians who will note the difficulties of correctly measuring the noise emission of a

rotor close to a hard reflecting surface. The loading on the motors is likely to be different to the in-flight conditions and this research aims to assess the suitability of this standard for the measurement of UAV sound power through comparisons with fly-over tests.

The UAV used for physical testing was a small videographer UAV, the HolyStone HS720 model. This model has 4 propellers, a MTOM of 630 g which fits within a footprint of 434 × 434 × 151 mm. It has a flight time of 22 min on a full battery and has a maximum control range of 3280 ft (1000 m). The sound power levels in each of the third octave bands were measured using a calibrated Svantek 971 sound level meter. Ten measurement points with duration 7 s each were completed. Tests were conducted indoors in a non-anechoic but acoustically treated space. A large rectangular room of dimensions 7.44 × 5.81 × 2.74 m which has previously been used for acoustic testing was used in this study [32]. Four Clearsonic S2466x2 (S5-2D) sound absorbing panels, dimensions 1670 × 609 × 38 mm, were also added to the space. Tests were also conducted outdoors on a large smooth concrete covered area free of any obstacles.

Compliance of the test environments with the requirements of ISO 3744:2010 was assessed. This standard allows for the testing of a UAV in a room which is adequately isolated from background noise provided that a correction factor can be applied to allow for a limited contribution from the reverberant field to the sound pressures on the measurement surface. The standard also allows for testing on a flat outdoor area. The criteria for the background noise is that the time-averaged sound pressure level averaged over the microphone positions shall be at least 6 dB, and preferably more than 15 dB, below the corresponding uncorrected time-averaged sound pressure level of the noise source under test when measured in the presence of the background noise.

The suitability of the indoor space for the measurement was verified through the use of a calibrated reference sound source, an Acculab RSS-101. The reference source was used to determine any required correction factors for each octave band due to the reverberant field during the indoor tests and to validate both sets of measurements. During the indoor tests the criteria for a 6 dB increase above the background was met for all frequency bands from 125 Hz upwards. The criteria for a 15 dB increase above the background was met for all for all frequency bands from 250 Hz upwards. For the outdoors tests the lower frequencies were more strongly affected by background noise with the criteria for a 6 dB increase above the background only being met above 500 Hz. Therefore it was decided to proceed with the indoor test environment for the final measurements.

A hemispherical measurement surface with a radius of 1 m was used according to the measurement positions detailed in the standard and reported in Table 1. $L_{Aeq}$ SPL readings were taken and converted to sound power levels using Equation (2).

$$L_W = L_{P,avg} + 10log(S) \qquad (2)$$

where $10log(S)$ can be translated to $20Log(r) + 8$ where $r$ is the chosen radius (in this case $r$ = 1 m). The measured sound power level of the reference sound source agreed with calibration data to within 0.78 dB. Correction factors to account for the reverberant level in the third octave bands between 50 and 20,000 Hz had a mean of 1.3 dB and a standard deviation of 1.5 dB.

Following the successful validation of the measurement procedure and facilities the UAV sound power was measured. The total sound power level of the UAV was calculated to be 86.8 dB (A). This compares well to literature values of measurements on smaller drones which range from 72.1 to 82.4 dB (A) [33]. Figure 3 shows the third octave band sound power spectra measured for the UAV and Table 2 reports the individual third octave band values. The expected speed of the motors is approximately 5250 RPM which puts the fundamental BPF in the 160 Hz third octave band. This tone and the first harmonic at twice the BPF are clearly visible in Figure 3.

**Table 1.** ISO 3744:2010 Microphone measurement positions.

| Position | x (m) | y (m) | z (m) |
|---|---|---|---|
| 1 | −0.99 | 0 | 0.15 |
| 2 | 0.50 | −0.86 | 0.15 |
| 3 | 0.50 | 0.86 | 0.15 |
| 4 | −0.45 | 0.77 | 0.45 |
| 5 | −0.45 | −0.77 | 0.45 |
| 6 | 0.89 | 0 | 0.45 |
| 6 | 0.33 | 0.57 | 0.75 |
| 8 | −0.66 | 0 | 0.75 |
| 9 | 0.33 | −0.57 | 0.75 |
| 10 | 0 | 0 | 1 |

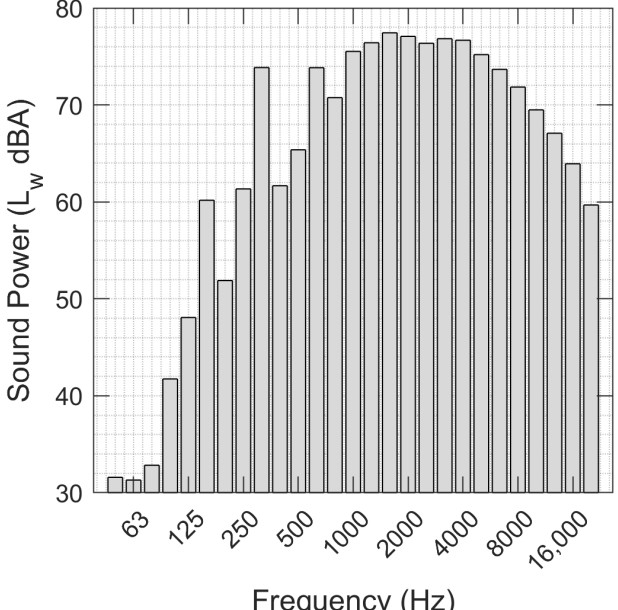

**Figure 3.** UAV Sound Power Measurement.

**Table 2.** UAV Sound Power Measurement.

| Frequency (Hz) | 50 | 63 | 80 | 100 | 125 | 160 | 200 |
|---|---|---|---|---|---|---|---|
| Lw (dB (A)) | 31.6 | 31.3 | 32.8 | 41.7 | 48.1 | 60.2 | 51.9 |
| Frequency (Hz) | 250 | 315 | 400 | 500 | 630 | 800 | 1000 |
| Lw (dB (A)) | 61.3 | 73.9 | 61.7 | 65.4 | 73.8 | 70.8 | 75.5 |
| Frequency (Hz) | 1250 | 1600 | 2000 | 2500 | 3150 | 4000 | 5000 |
| Lw (dB (A)) | 76.4 | 77.4 | 77.1 | 76.4 | 76.8 | 76.7 | 75.2 |
| Frequency (Hz) | 6300 | 8000 | 10,000 | 12,500 | 16,000 | 20,000 | **Total** |
| Lw (dB (A)) | 73.7 | 71.8 | 69.5 | 67.1 | 63.9 | 59.7 | **86.8** |

Table 3 shows the measured indoor directivity for the UAV, as measured at each of the ten microphone positions as described in annex B of ISO 3744:2010 and listed in Table 1. As can be seen from this table, the directivity of the UAV was seen to vary by over 2 dB across the angles tested. In this case the UAV is measured over a reflective plane which will cause a different directivity to the case of the UAV hovering in free space and may serve to reduce the magnitude of the measured directivity.

**Table 3.** Measured Directivity.

| Microphone Position | Measured Directivity dB |
|---|---|
| 1 | −1.00 |
| 2 | 0.69 |
| 3 | 0.16 |
| 4 | −0.74 |
| 5 | −1.11 |
| 6 | 0.58 |
| 7 | 0.27 |
| 8 | −0.65 |
| 9 | 2.38 |
| 10 | −2.45 |

## 4. Flyover Noise Measurements

The flyover tests were conducted on a sports ground within the university campus on an early weekend morning to avoid as much background noise as possible from traffic or activity from people within the grounds. Data was assessed during post-processing for evidence of background noise contamination. The size of the sports ground along with safety/security concerns for people within the vicinity resulted in the choice of 30 m as a maximum altitude for the UAV. The maximum height the UAV may fly is 120 m, according to Irish Aviation Authority regulations, however repeatable flight paths for light weight UAVs are a challenge at high altitudes due to wind.

The purpose of these measurements was to validate the noise contours at ground level produced by the modelling software. The flyover/hover tests were performed by mounting a Svantek 971 sound level meter on a fixed position at ground level at the centre of the sports field. While it is standard to mount a microphone at a height of 1.5 m above the ground this adds complications when comparing with the calculated noise contours, the use of a 1.5 m height would have required adding additional receiver positions into the modelling software which would be impractical for the variety of measurement locations considered. An additional disadvantage of the 1.5 m height is that it adds the potential for wind noise to contaminate the measurement. Authors have detailed the advantages of mounting the microphone at ground level for wind farm noise measurements [34,35]. The UAV was then flown over the microphone at certain heights & angles to measure the noise emission. For the flyover tests the sound level meter was set up to record in 100 ms segments. The instantaneous sound pressure values are required to generate the flyover profile. For the hover tests a long duration $L_{Aeq}$ value can be measured as the UAV is continuously operating at a fixed position. The Holy Stone HS270 is equipped with a GPS system that allows the user to monitor the above ground level (AGL). The heights tested were 5 m, 10 m, & 30 m. The angles tested at were 90°, 60°, & 45° with reference to the horizontal axis. These positions are shown in Figure 4.

Table 4 reports the overall sound pressure levels produced by the UAV at the various heights and angles tested. Since the straight line distance from the microphone position is not equal for the sideline tests, a distance correction assuming spherical wave spreading has been applied. This allows a direct comparison to the 90° results. The background noise at the measurement site was 45 dB (A). There is evidence of a measurable directivity under flight conditions that is greater than suggested by the indoor sound power measurements. There is a change of as much as 5 dB between the 90° and 45° measurement positions at 10 m height. This highlights a potential shortcoming in the use of ISO 3744:2010 as the standard measurement procedure for UAV noise within the EU.

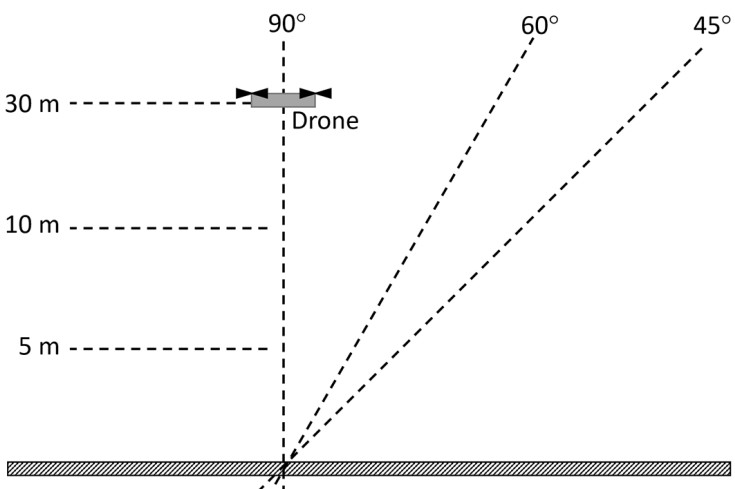

**Figure 4.** Hover test UAV positions.

**Table 4.** Hover tests $L_{Aeq}$ (dB (A)) as a function of distance and angle.

|       | 90°  | 60°  | 60° Corrected | 45°  | 45° Corrected |
|-------|------|------|---------------|------|---------------|
| 5 m   | 63.3 | 61.3 | 62.6          | 58.8 | 60.2          |
| 10 m  | 58.1 | 54.4 | 55.4          | 51.7 | 53.5          |
| 30 m  | 50.2 | 48.4 | 49.4          | 48   | 49.8          |

Figures 5 and 6 displays the third octave band results of two hover tests at 90° directly above the receiver and at distances of 5 m and 30 m. Figure 5 shows clear BPF tones including a third harmonic not as clearly present in the sound power tests. It is likely that the rotors experience more uneven loading during flight when compared to the indoor sound power measurements and this may increase the prominence of the BPF tones during in-flight noise measurements. These tones are still clearly present in the measurement at 30 m shown in Figure 6 although they have shifted to a slightly lower frequency band since the RPM of the UAV cannot be precisely controlled during flight.

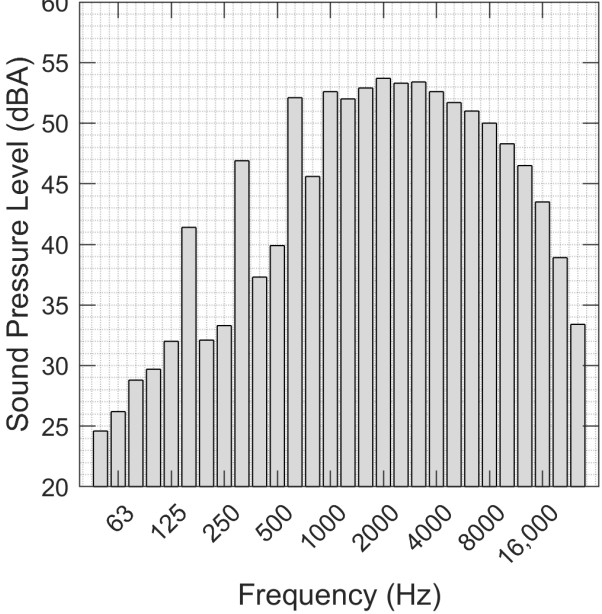

**Figure 5.** Hover tests—third octave band results 5 m 90°.

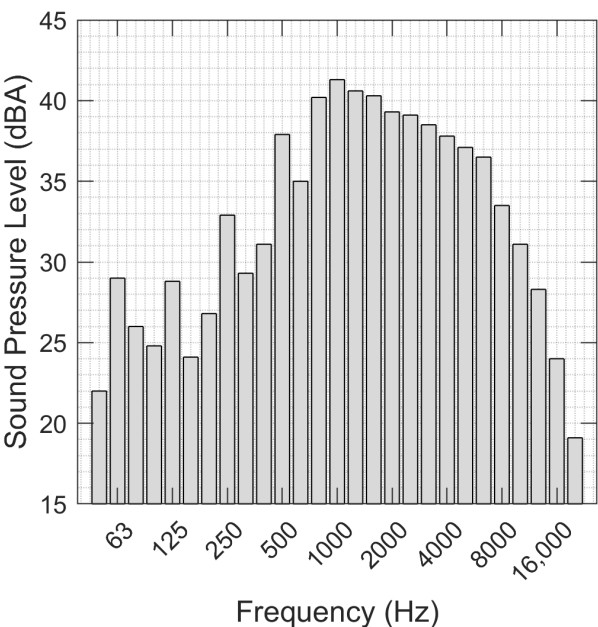

**Figure 6.** Hover tests—third octave band results 30 m 90°.

*Model Results*

The commercial implementation of ISO 9613 was used to model the hover tests. These were computed at the same positions as Figure 4. The sound power level as measured indoors was used as the sound power level of the sound source in the model. Due to the complexity of relating the indoor directivity measurements to hovering directivity, it was decided to assume an omnidirectional source. The modelling software requires directivity for every 10° increment from 0° to 180°, with directivity assumed to be rotational symmetric about the direction of emission. The determination of directivity as described in ISO 3744 had predetermined locations for the microphone positions which meant that in order to calculate the directivity at each of the 10° increments, interpolation would be required. Due to the relatively sparse measurement locations specified in ISO 3744 it was determined that any interpolation would not capture the required directivity. As a result, the calculated omnidirectional SPL was compared to experimental values with consideration given to the measured directivity to determine the accuracy of the model.

Table 5 shows a comparison of the modelled sound pressure level against the measured sound pressure level at various hovering conditions. The modelled results include UAV noise only and the UAV plus background noise results, where the background noise was 45 dB (A). As can be seen from the table, when the background noise was considered along with the modelled UAV noise, the overall sound pressure level as calculated by the software implementing IS0 9613, is within ±2.1 decibels in all instances. At distances beyond 10 m the UAV only noise drops below the level of the background noise leading to greater uncertainty in the experimental results.

**Table 5.** Modelled vs. measured sound power levels.

| Altitude | Degrees | Modelled SPL | | Measured SPL |
|---|---|---|---|---|
| - | - | UAV Only | UAV + Background | - |
| m | - | dB (A) | dB (A) | dB (A) |
| | 45 | 58.6 | 58.8 | 58.8 |
| 5 | 60 | 60.4 | 60.5 | 61.3 |
| | 90 | 61.7 | 61.8 | 63.3 |
| | 45 | 52.4 | 53.1 | 51.7 |
| 10 | 60 | 54.2 | 54.7 | 54.4 |
| | 90 | 55.6 | 56 | 58.1 |
| | 45 | 42.3 | 46.9 | 48 |
| 30 | 60 | 44.2 | 47.6 | 48.4 |
| | 90 | 45.7 | 48.4 | 50.2 |

Modelling the flyover tests involved modelling stationary UAVs at set distances from the receiver and recording the sound pressure level at the microphone for each of the corresponding distances. Source positions were set in 2 m intervals starting 30 m south of the receiver and ending 30 m north of the receiver. The calculated sound pressure levels where then plotted against the distance from the microphone. The results of one of the modelled flyover tests can be seen in Figure 7. These results assume a 5 m altitude above the receiver, at an angle of 90°, and again assume an omnidirectional source. The results of one of the experimental flyover tests can be seen in Figure 8. In this case the precise speed and location of the UAV could not be measured mid-flight so the plot of the flyover is presented as a function of time rather than distance from receiver. Validation of the UAV noise emission at high altitudes was challenging as the UAV is subject to higher wind speeds and uneven loading. The match between experimental and numerical values worsened as as altitude increased. The measured and modelled flyover profiles are in qualitative agreement suggesting that the assumption of omnidirectionality is reasonable.

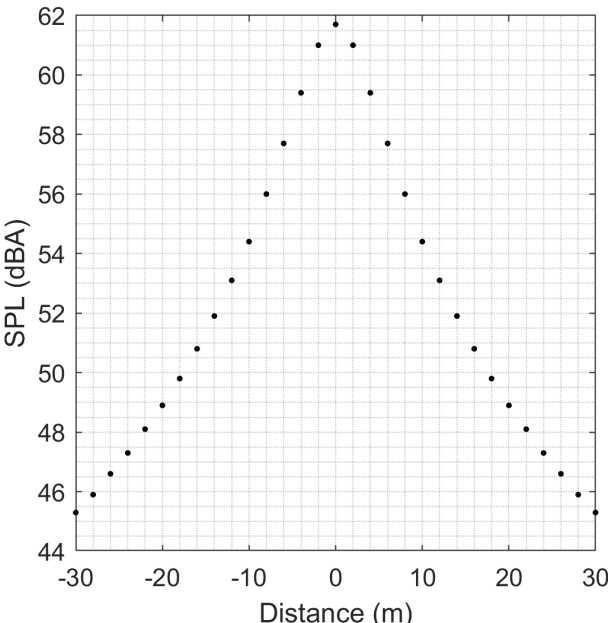

**Figure 7.** Modelled flyover test (5 m altitude, 90° angle).

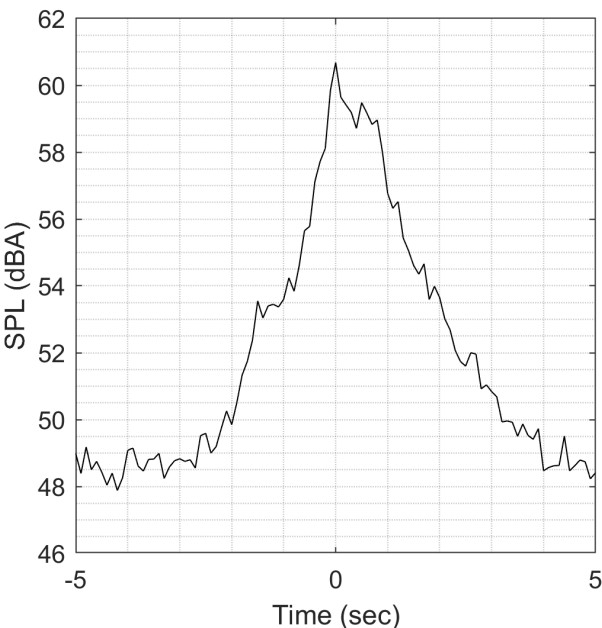

**Figure 8.** Measured flyover test (5 m altitude, 90° angle).

## 5. Case Study: Urban UAV Usage

The excellent agreement between the measured and modelled results validated the modelling approach for use in a wider case study. To understand the effect of the surroundings on the calculated sound pressure level, another model was created in a built up urban area. The location chosen was Grand Canal Dock in Co. Dublin, Ireland. This location contains offices and corporate headquarters as well as several restaurants, coffee shops and supermarkets. There are also many apartments and a hotel. The mixture of high rise buildings, businesses likely to employ UAVs in the future and the nearby residential buildings made this location a suitable one for the case study. In addition to the previous receiver location a second receiver was set up outside one of the apartment buildings, 10 cm from the facade of the building and the same hovering UAV positions as before where modelled. The location is shown in Figure 9.

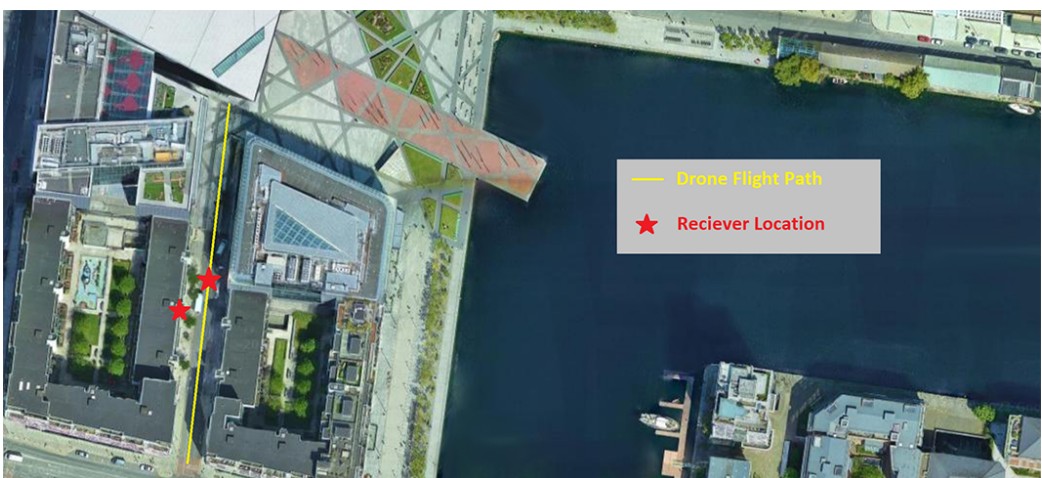

**Figure 9.** Layout of the case study site location 53.343253 N, −6.239823 W.

Data obtained from Dublin City Council reported the $L_{day}$ value for this location as 55–59 dB (A) as of 2017, while the $L_{DEN}$ was in the 60–64 dB (A) range [36]. The background noise assumed for this model was therefore chosen as 55 dB (A). The results can be seen in Table 6.

**Table 6.** Grand Canal Dock model results

| Altitude | Degrees | Modelled SPL | |
| --- | --- | --- | --- |
| - | - | UAV Only | UAV + Background |
| m | - | dB (A) | dB (A) |
| | 45 | 61.8 | 62.6 |
| | 60 | 63.8 | 64.3 |
| | 90 | 64.8 | 65.2 |
| | 45 | 56.4 | 58.8 |
| | 60 | 57.8 | 59.6 |
| | 90 | 58.9 | 60.4 |
| | 45 | 47.7 | 55.7 |
| | 60 | 47.4 | 55.7 |
| | 90 | 49.8 | 56.1 |

A final set of models were created to understand the maximum number of fly overs that could occur in this location before exceeding the World Health Organisation's maximum allowable $L_{DEN}$ for road traffic noise. In this case all other noise sources were excluded from the calculations. The UAVs flight path was modelled as a straight line path passing over a road in between two rows of buildings. The buildings heights varied between 12 m and 34 m, (however most of the buildings along the route had a height of less than 15 m) and their purposes ranged from shops to offices to apartments. Receivers where positioned 10 cm away from the outside walls of the buildings. Several flight altitudes where modelled to determine the effect of the UAVs altitude on the recorded $L_{DEN}$ values. The velocity used was the same as the one used in the physical flyover tests, 7.5 m/s. It was assumed that there were daytime flights only. As the use of commercial UAVs is still a relatively new area, there is currently no WHO recommended maximum $L_{DEN}$ for noise of this type. The road traffic maximum is therefore taken as the maximum for UAV noise, this value is 53 dB (A) [15]. Results from the flyover noise maps are shown in Figure 10 for heights of 5 m and 30 m. The total number of flights that can occur without exceeding WHO guidelines can be seen in Table 7. These results are relative to a background noise of 0 dB (A) and exclude all other existing noise sources such as road traffic.

**Table 7.** Maximum allowable fly overs for an $L_{DEN}$ of 53 dB (A).

| Altitude (m) | 5 | 10 | 20 | 30 |
| --- | --- | --- | --- | --- |
| Max flyby's allowed | 4600 | 5500 | 11,200 | 19,000 |

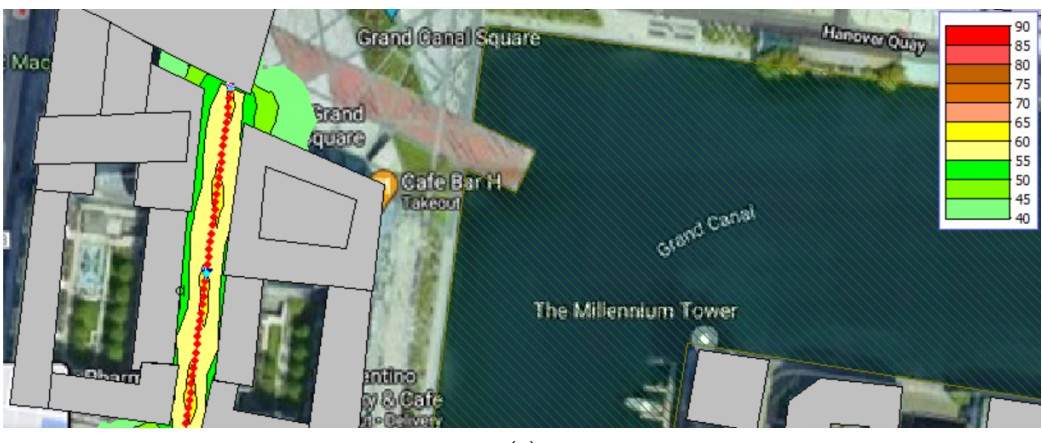

(**a**)

**Figure 10.** *Cont.*

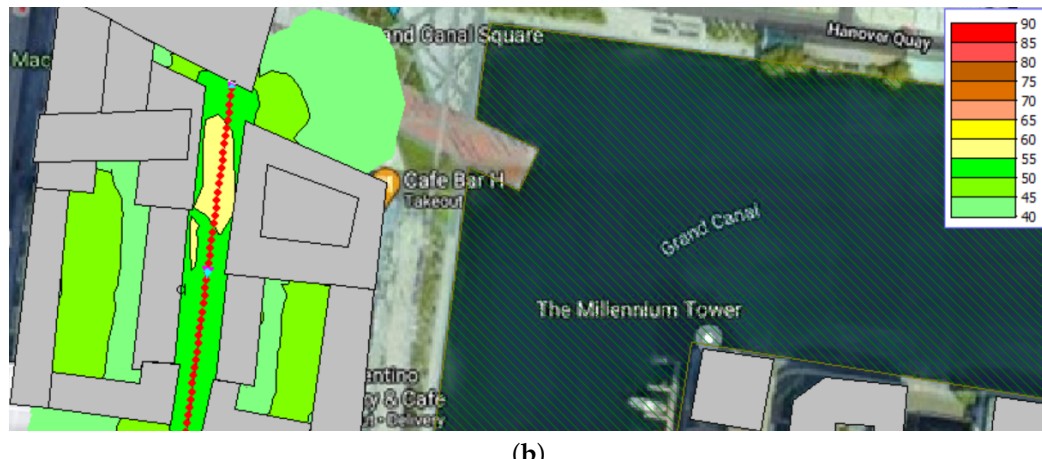

(**b**)

**Figure 10.** Fly over noise maps (**a**) 5 m (**b**) 30 m.

## 6. Discussion

The UAV used in this study was an entry level UAV used by both hobbyists and professional videographers. The sound power was measured in compliance with ISO 3744, the designated standard for estimating UAV sound power. The results demonstrate an acceptable level of accuracy for the sound power measurement can be achieved in the absence of an anechoic test facility. The UAV sound power was estimated as 86.8 dB (A) demonstrating the potential for even small scale UAVs to cause a considerable environmental noise problem. The current EU regulations for this class of UAV specify the maximum sound power as 84 dB (A) with a reduction four years after entry into force to 81 dB (A). The sound power measurements did contain strong tonal components at the blade passage frequencies. These tonal components were more clearly detected in noise measurements made of the UAV in flight with levels 10 dB in excess of adjacent octave bands. This indicates that real world operating conditions have a significant impact on the UAV noise emission and which should be considered when using ISO 3744 for estimating the sound power. The third octave results also indicate strong high frequency noise emission above 1000 Hz which may lead to increased noise annoyance.

Despite this limitation on the sound power measurements excellent results were achieved when modelling overall sound pressure levels using a commercial implementation of ISO 9613. Hovering tests at a range of distances and angles agreed within ±2.1 decibels when experimental background noise is accounted for. From the measured directivity of the UAV, as seen in Table 3, the directivity of the UAV varies within ±2.4 dB. It is possible that improved estimation of the UAV noise directivity may increase the accuracy of the modelled results. Where sound power testing is conducted over a hard reflective surface the true directivity of the UAV noise source is difficult to assess.

While not ideally suited to modelling an individual flyover measurement the software gave good qualitative agreement with the experimental flyover profile. This indicates that the assumption of omnidirectionality may be suitable in the first instance when modelling UAV noise emission.

From the modelled results, it can be seen that the effect of background noise is most apparent at greater distances from the receiver (microphone). At distances of 30 m the UAV noise was comparable to the background noise at the test site. In these cases, the addition of background noise increased the overall sound pressure level calculated at the receiver by over 4 dB. This also is in agreement with the findings of Torija et al [17] who stated that louder background noise reduces the impact of UAV noise.

The modelling tool was used in a case study of potential UAV use considering a fleet of equivalent UAVs operating in an urban setting. The Grand Canal dock area of Dublin was chosen due to the mix of uses and the availability of background noise data. With a background noise level of $L_{day} = 55$–59 dB (A) and $L_{DEN} = 60$–64 dB (A) this site already

exceeds WHO guidelines for the maximum $L_{DEN}$ value for road traffic. The reflective surfaces of the surrounding buildings led to increased UAV noise due to reflections (Table 5 vs. Table 6). For the range of UAV positions investigated previously a stationary hovering UAV produced levels between 47.4–64.8 dB (A), well in excess of the $L_{day}$ value when the UAV is below a height of 10 m.

At the lower altitudes the SPL was approximately 3 dB higher in each case in Grand Canal Dock than the original model, which was to be expected as the sound is reflected off of the surrounding building facades. The addition of the background noise also had a measurable effect when determining the overall sound pressure level. From the data in Table 6, the effect of the UAV noise is more apparent at lower altitudes. At each of the positions at an altitude of 30 m, the UAV noise had relatively little effect on the overall sound pressure level. At low altitudes, even with high background noise such as this case, the addition of the UAV noise has a large effect on the predicted sound pressure level. The total SPL was increased in some cases by over 10 dB compared to the background noise only. An increase of this magnitude would likely cause significant annoyance in the areas surrounding the UAV, as demonstrated by Palmer et al. [23].

The models to determine the maximum number of flyovers permitted to remain within the maximum $L_{DEN}$ for road traffic noise showed the effect of the buildings on the recorded values. When excluding all other noise sources the maximum number of flyovers was high for each of the altitudes modelled. For flight altitudes below the heights of the majority of the buildings far fewer flights were possible when compared to the altitudes above the buildings (5500 flights for 10 m vs. 19,000 flights for 30 m). This is partly due to the source being closer to the receiver, but also due to the sound reflecting from the facades of the buildings, increasing the perceived noise at the receive locations. It is important to note that as the noise limit is for road traffic only, these models did not include the measured background noise for the location, as their purpose was to investigate the UAV noise only. These models demonstrate the importance of flying at altitude, as it would allow more UAV flights over the area, while also mitigating the noise experienced in noise sensitive areas such as residential and office locations below the flight path.

## 7. Conclusions

The study successfully modelled and experimentally validated the noise emission from a small scale UAV. This has validated the use of ISO 3744 for sound power assessments of UAVs. The UAV used in this work was a class C1 UAV according to EU legislation. The sound power of a C1 UAV was limited to 85 dB (A) from entry into force of the EU regulations and the UAV used in this work already exceeds this value at 86.8 dB (A). This highlights the significant work ahead for the industry to meet the current and upcoming legislative requirements for operating UAVs in the European Union. The data produced by the upcoming regulation of UAV noise emission within the European Union can provide the necessary input for UAV environmental noise assessments. The key conclusions are:

- While there are issues with the use of ISO 3744 for the measurement of UAV sound power in terms of directivity assessments the good match between measured and modelled fly-over noise levels suggests that it is suitable for UAV noise assessments.
- The UAV sound power can be accurately measured in compliance with ISO 3744 in non-anechoic test facilities provided the correct background noise assessments and reverberant field correction factors are applied. This opens up low cost testing for the wide variety of commercial UAVs currently available and enables companies to meet the requirements of EU legislation.
- The BPF tones were more prominent during in flight testing than the sound power measurements. ISO 3744 may fail to take into account all of the features present during in-flight noise emission of UAVs.
- The sound power of the UAV used exceeded currently EU limits by 1.8 dB (A) demonstrating the potential for even small UAVs to cause serious noise issues.

- Considering the A-weighted in-flight noise emission the noise emitted by the UAV is mainly composed of higher frequencies above 1000 Hz which may increase annoyance.
- The measured sound power provided accurate estimates of UAV noise emission when used as an input to commercial implementations of ISO 9613.
- Low cost, indoor measurements of sound power level are sufficient to develop models and noise maps for UAVs operating in a variety of conditions.
- Assuming an omnidirectional source to simplify calculations when creating noise models can still provide accurate results. The differences observed between the measured and modelled results were of the same magnitude as the measured directivity suggesting that including directivity effects may resolve the remaining differences.
- When considered in addition to existing noise sources in an urban setting UAV noise is likely to significantly exceed WHO guidelines.

**Author Contributions:** Conceptualization, J.K., K.C. and S.G.; methodology, J.K., K.C. and S.G.; software, K.C.; validation, S.G.; formal analysis, J.K., K.C. and S.G.; investigation, J.K., K.C. and S.G.; resources, J.K.; data curation, J.K.; writing—original draft preparation, K.C.; writing—review and editing, J.K., K.C. and S.G.; visualization, J.K., K.C. and S.G.; supervision, J.K.; project administration, J.K.; funding acquisition, J.K. All authors have read and agreed to the published version of the manuscript.

**Funding:** This research received no external funding.

**Informed Consent Statement:** Not applicable.

**Data Availability Statement:** The data required to reproduce this work is included in the paper.

**Conflicts of Interest:** The authors declare no conflict of interest.

## Abbreviations

The following abbreviations are used in this manuscript:

| | |
|---|---|
| UAV | Unmanned aerial vehicle |
| eVTOL | electric vertical takeoff and landing |
| BPF | Blade passage frequency |
| MTOM | Maximum take off mass |
| SPL | Sound pressure level |

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
