# Peer review of "UAV Noise Emission—A Combined Experimental and Numerical Assessment"

_acoustics, doi:10.3390/acoustics4020018_

Round 1

Reviewer 1 Report

The paper can represents a fine addition to the acoustic emission of UAVs field. The studied arguments are wide, but not too deeply investigated. I suggest the authors to enrich the little details, before going into large arguments. I would say that a proper definition of the work would have classified and studied noise emission of many UAVs, leading to a proper publication. Then, a second work would have studied the fly over, and the noise map. At present, it is an ensemble of “little things” that would not represent a particular innovation, but still something publishable if many aspects will be addressed.

Avoid brands in favor of generic sentences.

The part dealing with health effect of noise is poor and should be improved. A suggestion can be a more detailed period, and references, like: “Exposure to noise is associated to sleep disorders with awakenings (Muzet A. Environmental noise, sleep and health. Sleep Med Rev 2007; 11: 135–42), learning impairment (Zacarías, F. F., Molina, R. H., Ancela, J. L. C., López, S. L., & Ojembarrena, A. A. (2013). Noise exposure in preterm infants treated with respiratory support using neonatal helmets. Acta Acustica united with Acustica, 99(4), 590-597; Erickson, Lucy C., and Rochelle S. Newman. "Influences of background noise on infants and children." Current Directions in Psychological Science 26.5 (2017): 451-457.), hypertension ischemic heart disease (Dratva, J., et al. (2012). “Transportation noise and blood pressure in a population‐based sample of adults.” Environmental Health Perspectives, 120(1): 50–55. Babisch, W., Beule, B., Schust, M., Kersten, N., Ising, H., ‘Traffic noise and risk of myocardial infarction’, Epidemiology, 16, 2005, pp. 33–40. ), diastolic blood pressure (Petri, D., Licitra, G., Vigotti, M. A. & Fredianelli, L. (2021). Effects of Exposure to Road, Railway, Airport and Recreational Noise on Blood Pressure and Hypertension. Int. J. Environ. Res. Public Health 2021, 18(17), 9145), reduction of working performance (Vukić, L., Fredianelli, L., & Plazibat, V. (2021). Seafarers’ Perception and Attitudes towards Noise Emission on Board Ships. International Journal of Environmental Research and Public Health, 18(12), 6671. Rossi, L., Prato, A., Lesina, L., & Schiavi, A. (2018). Effects of low-frequency noise on human cognitive performances in laboratory. Building Acoustics, 25(1), 17-33.), annoyance (Miedema HME, Oudshoorn CGM. Annoyance from transportation noise: relationships with exposure metrics DNL and DENL and their confidence intervals. Environ Health Perspect 2001; 109: 409–16).”

The last part of introduction is not correctly introducing what the paper will do, and how it will be done.

Fig. 1 report also broadband level.

I failed to understand how many UAVs have been measured. This will let me understand the statistic behind the results of noise emission characterization and see if the result is generalizable or not.

Put a space after digits and before units.

How the blades rotational speed is considered in the noise emission? I can imagine that there will be an important dependence and that a single value would not be enough if not properly related to an rpm.

I do not agree with placing the mic at ground, as it is not representative for citizens heigh when performing noise measurements.

Report a sound pressure level time history of typical pass-by of a UAV.

What was the height of pass-by? Please also discuss the effect that different heights have on the model.

Line 285. This assumption is definitively wrong. There are many references in literature showing how different source have different impact on health, especially annoyance and sleep disorders, and noise limits would reflect so. For Road traffic noise and wind turbine noise see (Fredianelli, L., Carpita, S., & Licitra, G. (2019). A procedure for deriving wind turbine noise limits by taking into account annoyance. Science of the total environment, 648, 728-736) but also consider references for airplane noise, and more (Janssen, Sabine A., et al. "Exposure-response relationships for annoyance by wind turbine noise: a comparison with other stationary sources." EURONOISE, 8th European conference on noise control, Edinburgh, Scotland, United Kingdom, 26–28 October 2009. Institute of Acoustics, 2009.). Sure, UAVs need a proper evaluation of annoyance, and so on.

Conclusions should better summarize what the paper has investigated, before reporting the real conclusions. This is a suggestion made for lazy readers, as there should be. In fact, many potential readers would only read abstract and conclusions. Even if it is not a proper way of reading papers, authors should maximize the potential catch of readers giving them more info in the conlusions.

Reviewer 2 Report

In some measurements of the sound level of drones (see Iannace, Ciaburro) there is a tonal composition at 4kHz. I don't see it in your paper.
I see high frequency tones, but only low frequency ones.
You should add a photo of the drone.
You should add a photo of the propellers with the dimensions specify the number of revolutions of the motors.
You should add a photo during the field measurements.
The noise of the drones a few tens of meters is not perceptible. Why do you say that drone noise is annoying in traffic?
Have you asked people what they think about this noise with interviews?
You could run tests in the lab with the noise of the drone and make people listen to it.
Did you use commercial iNoise software, how did you build the model?
Have you compared the values ​​of the sound power level Lw with that of other authors? 
Please see: 
Papa, U et al."Sound power level and sound pressure level characterization of a small unmanned aircraft system during flight operations". Noise and Vibration Worldwide, 2017

Round 2

Reviewer 1 Report

The authors did a massive work in the revision phase and the paper is now ready for being published.

Reviewer 2 Report

accept